# Comprehensive transcriptomic analysis of long non-coding RNAs in bovine ovarian follicles and early embryos

**Pengmin Wang, Éric R. Paquet\*, Claude Robert\***

Département des sciences animales, Faculté des sciences de l'agriculture et de l'alimentation, Université Laval, Québec City, Québec, Canada

\* eric.paquet@fsaa.ulaval.ca (ERP); claude.robert@fsaa.ulaval.ca (CR)

**Data Availability Statement:** This manuscript is a comprehensive re-analysis of datasets available at NCBI. All dataset are published and freely available.

**Funding:** The authors acknowledge the following funding agencies for their support: Natural

## Abstract

Long non-coding RNAs (lncRNAs) have been the subject of numerous studies over the past decade. First thought to come from aberrant transcriptional events, lncRNAs are now considered a crucial component of the genome with roles in multiple cellular functions. However, the functional annotation and characterization of bovine lncRNAs during early development remain limited. In this comprehensive analysis, we review lncRNAs expression in bovine ovarian follicles and early embryos, based on a unique database comprising 468 microarray hybridizations from a single platform designed to target 7,724 lncRNA transcripts, of which 5,272 are intergenic (lincRNA), 958 are intronic, and 1,524 are antisense (lncNAT). Compared to translated mRNA, lncRNAs have been shown to be more tissue-specific and expressed in low copy numbers. This analysis revealed that protein-coding genes and lncRNAs are both expressed more in oocytes. Differences between the oocyte and the 2-cell embryo are also more apparent in terms of lncRNAs than mRNAs. Co-expression network analysis using WGCNA generated 25 modules with differing proportions of lncRNAs. The modules exhibiting a higher proportion of lncRNAs were found to be associated with fewer annotated mRNAs and housekeeping functions. Functional annotation of co-expressed mRNAs allowed attribution of lncRNAs to a wide array of key cellular events such as meiosis, translation initiation, immune response, and mitochondrial related functions. We thus provide evidence that lncRNAs play diverse physiological roles that are tissue-specific and associated with key cellular functions alongside mRNAs in bovine ovarian follicles and early embryos. This contributes to add lncRNAs as active molecules in the complex regulatory networks driving folliculogenesis, oogenesis and early embryogenesis all of which are necessary for reproductive success.

## Introduction

High-throughput genome sequencing technology has revealed that most transcribed RNA encodes little or no protein [1, 2]. These RNA transcripts are referred to as non-coding RNAs (ncRNAs), which may be housekeeping or regulatory. Housekeeping ncRNAs like

Sciences and Engineering Research Council of Canada (Grant RGPIN-2017–04775) and PW is supported by a scholarship from the Fonds Québécois de la Recherche sur la Nature et les Technologies. The funders had no role in study design, data collection and analysis, decision to publish, or preparation of the manuscript.

**Competing interests:** The authors have declared that no competing interests exist.

transfer RNAs (tRNAs), ribosomal RNAs (rRNAs), small nuclear RNAs (snRNAs), and small nucleolar RNAs (snoRNAs) are generally expressed constitutively and are required for normal cell function and viability. Regulatory ncRNAs are grouped into two broad classes based on length: short ncRNAs and long ncRNAs (lncRNAs). Short ncRNAs are less than 200 nucleotides in length and includes small interfering RNAs (siRNAs), microRNAs (miR-NAs) and piwi-interacting RNAs (piRNAs), which are usually highly conserved across species and are involved in transcriptional and post-transcriptional gene silencing [3]. LncRNAs exceed 200 nucleotides in length and have emerged as a major class of transcript [4]. In terms of biogenesis and structure, lncRNAs share many features with messenger RNAs, often transcribed by RNA polymerase II and spliced [5], bearing a 7-methylguanosine cap at the 5' end and a polyadenylated tail at the 3' end. On the other hand, lncRNAs have fewer but slightly longer exons, are expressed in lower copy numbers and with more cell and tissue specificity [1], have shorter half-lives [6], and are generally less conserved across species [7].

The more lncRNAs are studied, the more they are shown to be functional RNA molecules. For example, they have been implicated in regulation of gene expression through a variety of mechanisms such as regulation of transcription [8], alternative splicing [9], mRNA translation [10–12], and epigenetic regulation [13, 14]. LncRNAs are also involved in diverse cellular processes including proliferation [15], apoptosis [16], pluripotency and differentiation [17], and in embryonic growth and development [18, 19]. Current knowledge about lncRNAs comes from variety of animals, from mice to humans [20, 21].

Cattle provide a suitable model for studying ovarian function in humans since they are mono-ovulatory with a similar endocrine cycle. However, surprisingly few characterizations of lncRNAs in early embryonic development in cattle have been published [22–25].

Bovine folliculogenesis, oogenesis and early pre-hatching development are studied mainly to understand the succession of key events that include follicular support leading to the formation of a developmentally competent oocyte, fertilization, embryonic genome activation, and initiation of cell lineage differentiation. The study of early development is important for perfecting assisted reproductive technologies such as *in vitro* oocyte maturation and fertilization for embryo production and transfer [26]. RNA sequencing has shown that the transcriptome is more complex than expected [27]. While most microarrays are designed solely to study mRNAs and are blind to lncRNAs, the EmbryoGENE microarray platform was designed using a comprehensive transcript catalog derived from an RNA sequencing survey that included novel transcripts [28]. This microarray was used in concordance with standard operation procedures for sample preparation, hybridization, and processing, making all independent studies that used it fully compatible. However, due to the absence of functional information associated with lncRNAs, downstream analyses were focused on mRNAs. This now provides a unique database to perform a comprehensive analysis to start filling this knowledge gap by describing the extent of their expression in bovine ovarian follicles and early embryos alongside expressed mRNAs.

Therefore, we assembled and curated a compendium of 468 transcriptomes from 47 different experimental conditions to characterize lncRNAs expression during bovine early development. These uncharacterized transcripts were classified according to their genomic position in relation to protein-coding genes. We then defined lncRNAs that are specifically associated with folliculogenesis, oogenesis and early embryogenesis, and analyzed the joint expression of lncRNAs and mRNAs across tissues to associate potential functions to lncRNAs based on co-expression networks.

## Materials and methods

### Reannotating, assembling and curating the datasets

The 43,794 probe sequences present on the EmbryoGENE microarray were downloaded from the website (http://emb-bioinfo.fsaa.ulaval.ca/) on April 2020 and were mapped on the ARS-UCD1.2/bosTau9 (Apr.2018) reference genome using BLAT with the following parameters: stepSize = 5, repMatch = 2253, minScore = 20, minIdentity = 0, and mismatch = 2 [29]. Of these, 34,780 (79%) were mapped on the latest genome. Using bovine genome annotations obtained from UCSC, Ensembl, and NCBI, probes were then reannotated separately, as mRNAs if they overlapped with at least one exon on the same strand, as intronic lncRNAs if they were within introns of protein-coding genes on the same strand, as antisense lncRNAs if they overlapped with protein-coding genes on the opposite strand, and finally, as intergenic lncRNAs if they did not overlap with protein-coding genes. Probes annotated ambiguously as mRNAs or lncRNAs in the same database were discarded. To lower the risk of errors in identifying lncRNAs, we kept 30,575 probes left after this annotation procedure and assigned them to lncRNAs if all three databases concurred. Only one database was required to assign a probe to mRNA. To reduce redundancy when a plurality of probes targeted the same gene, the probe with the highest mean signal was selected, which reduced the total to 21,840. The entire workflow leading to the final annotation is represented in Fig 1.

All raw microarray data associated with the EmbryoGENE platform (GPL13226) were retrieved from the NCBI Gene Expression Omnibus on August 2020 [30]. This platform uses standardized samples processing procedures. Tests were done during the platform development phase to minimize method-induced variance including parameters to extract total RNA,

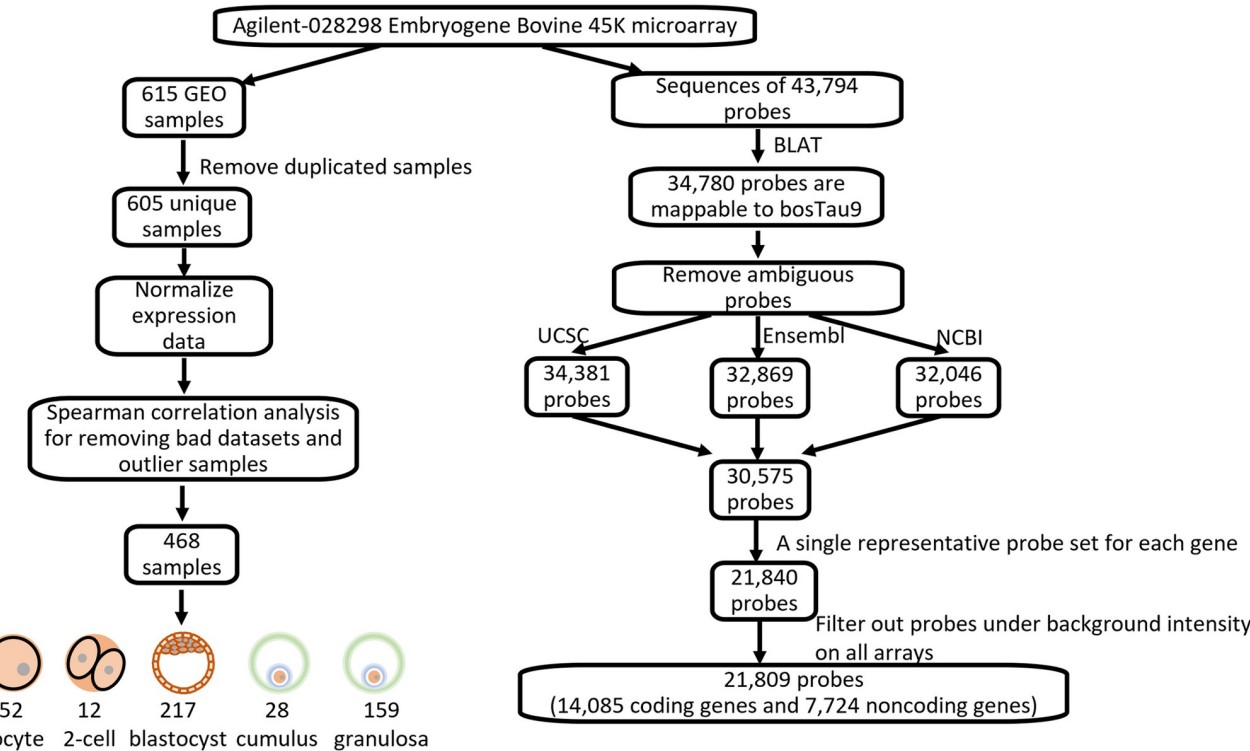

**Fig 1. Workflow by which the EmbryoGENE compendium of gene expression for this study was assembled.**

the global amplification step and the microarray hybridization conditions [31]. It has been used for 74 Gene Expression Omnibus series comprising 663 individual samples. Some of these 74 series are repeated and some are also listed on a different platform (GPL18384). We retained 605 unique samples from 62 expression series and used the signal from the Cy3 (green) channel for all downstream analyses. All raw signal intensities were $\log_2$ transformed and quantile normalization was applied across all data from all arrays to uniformize the distribution and thereby make all expression levels comparable [32]. Spearman correlation analysis then allowed us to note relationships between the different datasets and individual samples. The datasets and samples that did not cluster with the others were called outliers and removed from downstream analysis, which led to a final dataset of 468 samples (S1 Fig).

The EmbryoGENE platform is composed of 604 DarkCorner plus 153 structural negative probes (3xSLv1) usable as negative controls to measure if expression level is above the background. We considered a probe to be expressed on an array if its raw intensity exceeded the $95^{th}$ percentile of the raw intensities of these negative controls. Based on this cutoff, 31 probes were considered not expressed at all in any sample and were discarded from downstream analyses. We thus retained a final set of 14,085 probes targeting mRNA and 7,724 probes targeting lncRNA from 468 quantile-normalized samples from different embryo developmental stages.

## Weighted gene co-expression network analysis (WGCNA)

Network analysis was performed using R package WGCNA protocol version 1.70 [33]. We chose a soft thresholding power (β) of 12 to construct the network, based on the criterion of approximate scale-free topology $R^2$ = 0.9 and mean connectivity < 100 (S2A and S2B Fig). The co-expression modules were then constructed using automatic module detection with the blockwiseModules function as follows: β = 12, corType = "pearson", networkType = "signed", TOMType = "signed", minModuleSize = 30, maxBlockSize = 21809, mergeCutHeight = 0.25, and reassignThreshold = 0. Gene modules were distinguished by color and appeared as branches on the clustering tree. To identify tissue-related modules, the correlation between tissue type (each tissue type as a column and each sample as a row with binary variables) and module eigengenes (1st principal component explaining most variation in the module expression levels) was calculated using Pearson correlation. We also identified potential transcription factors present in individual modules using the AnimalTFDB3.0 database [34].

## Functional enrichment analysis

To elucidate the putative biological functions of lncRNAs in each module, mRNAs annotated by the enricher function in R package clusterProfiler [35] version 4.4.4 were analyzed using Gene Ontology enrichment and Kyoto Encyclopedia of Genes and Genomes (KEGG) downloaded from http://geneontology.org/ (goa_cow.gaf, last updated July 1, 2022) and https://www.genome.jp/kegg/ (bta00001.keg, last updated July 17, 2022) respectively. Pathway analysis was restricted to metabolism, genetic information processing, environmental information processing, cellular processes, and organismal systems. The enricher function uses a hypergeometric test to assess significance and we used a Benjamini-Hochberg adjusted p-value < 0.05 to select enriched functions associated with individual modules.

## Construction of lncRNA-mRNA-TF co-expression networks and finding hub transcription factors and hub genes

A Pearson correlation coefficient and p-value were calculated for each mRNA/lncRNA pair of transcripts in modules using the corAndPvalue function in the R package WGCNA. Pairs with correlation coefficient > 0.7 and p-value < 0.05 were selected to build the correlation network

and identify the transcription factors and genes that had the largest correlation coefficient sums were designated as pathway hubs.

## Results

### Assembling a large compendium of bovine early development RNA abundance data

The following analysis enabled to characterize the expression of lncRNAs in the bovine ovarian follicle and early embryos. Data gathered over several years from the EmbryoGENE gene expression platform allowed us to assemble a curated compendium of 468 transcriptomes obtained from ovarian cells and early bovine development. Briefly, this platform was created to study coding and non-coding gene expression in the context of bovine early development. To create this platform, an exhaustive survey of all RNA species present during folliculogenesis and bovine early development was first performed using total RNA-seq and subsequently used to design a dedicated array and standardized protocols to be used by all consortium researchers [28].

All probes listed on this platform were reannotated using the most recent bovine genome ARS-UCD1.2/bosTau9 (Apr.2018)). A total of 14,085 probes target mRNAs and 7,724 probes target lncRNAs. Based on proximity to nearby protein-coding genes, lncRNAs were subdivided into 5,272 intergenic (lincRNAs, 68%) and 2,452 genic lncRNAs (32%) probes. LincRNAs were further categorized using the distance and the strand of their nearest genes. About 9% of them might be products of gene 5' or 3' UTR extensions since they are located within 1 kb on the same strand. However, most lincRNAs (82%) are located more than 1 kb from known protein-coding genes and 21% are located more than 50 kb away (S1 Table).

Our study of the probes distribution throughout the bovine genome indicates that the reannotated compendium covers most chromosomes uniformly except for 18, 19, and 25 that have higher densities of both lncRNAs and mRNAs (Fig 2). The number of lncRNAs per chromosome is strongly correlated with mRNAs (Pearson's $\rho$ = 0.89, p-value = $3.4e^{-11}$). Furthermore, we also performed several quality controls of the data used in the dataset and performed a global normalization of the dataset to be able to jointly analyze the data (see Material and methods). Given the coverage by the probes, the dataset size and spanning of different bovine developmental stages, this assembled and curated compendium is a good resource to characterize the joint gene expression profile of mRNAs and lncRNAs.

### Developmental stage specificity of mRNAs, genic lncRNAs and lincRNAs

Gene expression during early development is quite heterogenous and dynamic and progresses in stages [36]. To understand how lncRNAs and mRNAs expression patterns might also be stage specific, we performed multiple principal component analyses using different proportions of mRNAs, genic lncRNAs, and lincRNAs (Fig 3). Since mRNAs, genic lncRNAs and lincRNAs expression profiles differ in numbers, the three classes were balanced beforehand to ensure equal contribution. A PCA combining mRNAs and lncRNAs distinguishes between the early developmental stages (oocyte through 2-cell), the blastocyst or somatic, cumulus, and granulosa cells (PC1 in Fig 3A). If mRNAs alone are used, the oocyte and 2-cell stages still differ from the others (PC1 and PC2), but blastocysts now differ from cumulus and granulosa cells (PC2), suggesting that mRNAs are more specific than lncRNAs for characterizing embryonic development (Fig 3B). If only genic and lincRNAs are analyzed, the oocyte and 2-cell stages remain distinguishable whereas the other stages are harder to distinguish, suggesting

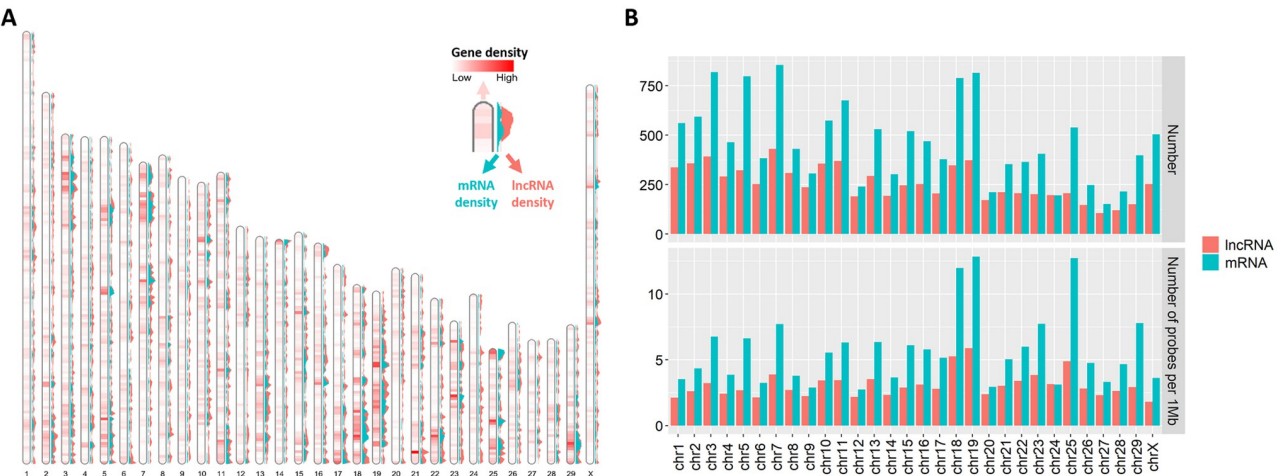

**Fig 2. Location of the 21,809 selected and reannotated mRNA and lncRNA probes on the bovine genome.** (A) Probe distribution and approximate abundance (color intensity) on each chromosome based on the Refseq annotation from bosTau9. (B) Number of probes per chromosome and per $10^6$ (1Mb) nucleotides.

that lncRNAs expression is more distinctive for the early stages (Fig 3C–3E). Overall, this analysis reveals that lncRNAs abundance is a distinctive feature of early development at stages still under the control of maternal reserves.

## Tissue-wide and tissue-specific expressions of lncRNAs and mRNAs

Among the 468 samples in the dataset, 217 were from blastocysts, 159 from granulosa cells, 52 from oocytes, 28 from cumulus cells, and 12 from 2-cell embryos. Tissue-specific expression was identified for both lncRNAs and mRNAs. A probe was considered expressed in a tissue if detected in more than 65% of samples (Fig 4A). In general, oocytes expressed larger numbers of mRNAs and lncRNAs in comparison to cumulus, granulosa, 2-cell and blastocysts, which is consistent with the oocyte transcribing and storing of a large amount of maternal RNA needed during the early stages of development [37]. In terms of the number of transcripts expressed (Fig 4B), the tissues ranked as follows: oocyte (16,545) > cumulus (15,284) > granulosa (15,186) > blastocyst (14,623) > 2-cell (14,246). Fig 4B also shows that a small majority of transcripts (11,792 or 54%) detected by the microarray are expressed in all five tissues, and fewer are tissue-specific or expressed in only one tissue. The oocyte contains more tissue-specific transcripts than any other tissue and more than half of these (563/907) are lncRNAs, which is the highest proportion observed. Careful examination of the proportions of tissue-specific lncRNAs shows a decrease as development progresses.

The tissue distributions of lncRNAs and mRNAs expression are shown in Fig 4C. LncRNAs have a more tissue-specific pattern of expression while mRNAs tend to be expressed more broadly. About 63% of mRNAs were detected in all five tissues compared with 38% of lncRNAs while the opposite was observed when a single tissue is considered (5% of mRNAs versus 11% of lncRNAs).

In terms of overall expression level, lncRNAs were less abundant than mRNAs (Fig 4D–4F). We also note that most transcripts detected in all five tissues were highly expressed, whereas transcripts detected in only one tissue tended to be less expressed (Fig 4E and 4F). On the other hand, when considering only tissue-specific transcripts, lncRNAs and mRNAs were almost equally abundant (Fig 4F).

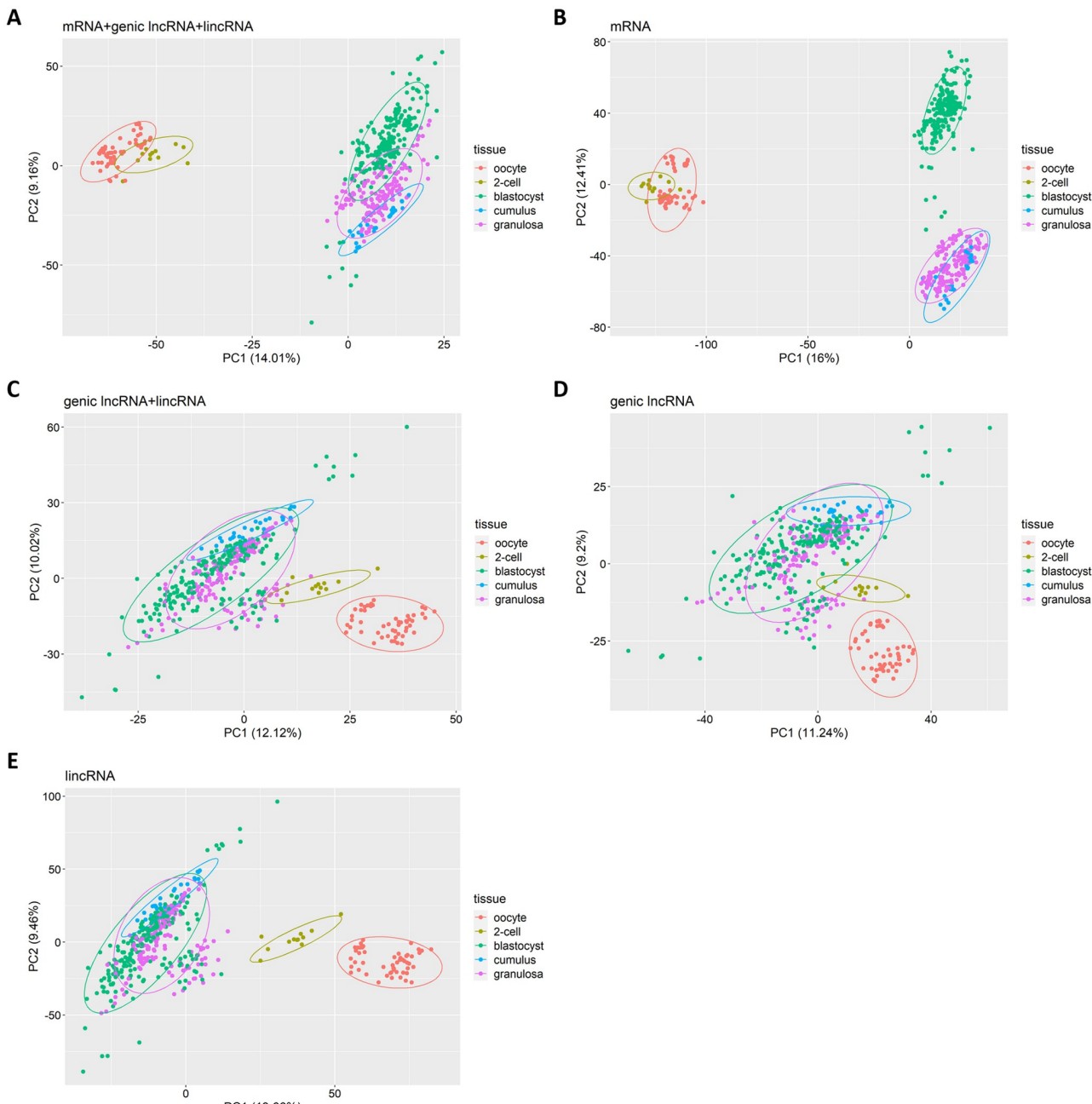

**Fig 3. Joint and individual principal component analyses of mRNAs, lincRNAs and genic lncRNAs expression in early developmental stages of bovine embryos.** (A) Of a random set of 2,000 mRNAs, 1,000 genic lncRNAs, and 1,000 lincRNAs. (B) Of all mRNAs (n = 14,085). (C) Of a random set of 1,000 genic lncRNAs and 1,000 lincRNAs. (D) Of all genic lncRNAs (n = 2,452). (E) Of all lincRNAs (n = 5,272). Ellipses contain 95% of the samples of the given cell or tissue type.

## LncRNAs and mRNAs co-expression analysis using WGCNA

As means of inferring indirectly the roles potentially played by lncRNAs in the ovary and in early embryos, co-expression analyses using WGCNA was performed on the profiles of the 14,085 mRNAs and 7,724 lncRNAs molecules detected by the microarray and 25 modules were obtained. The grey module contains all transcripts that could not be assigned to any

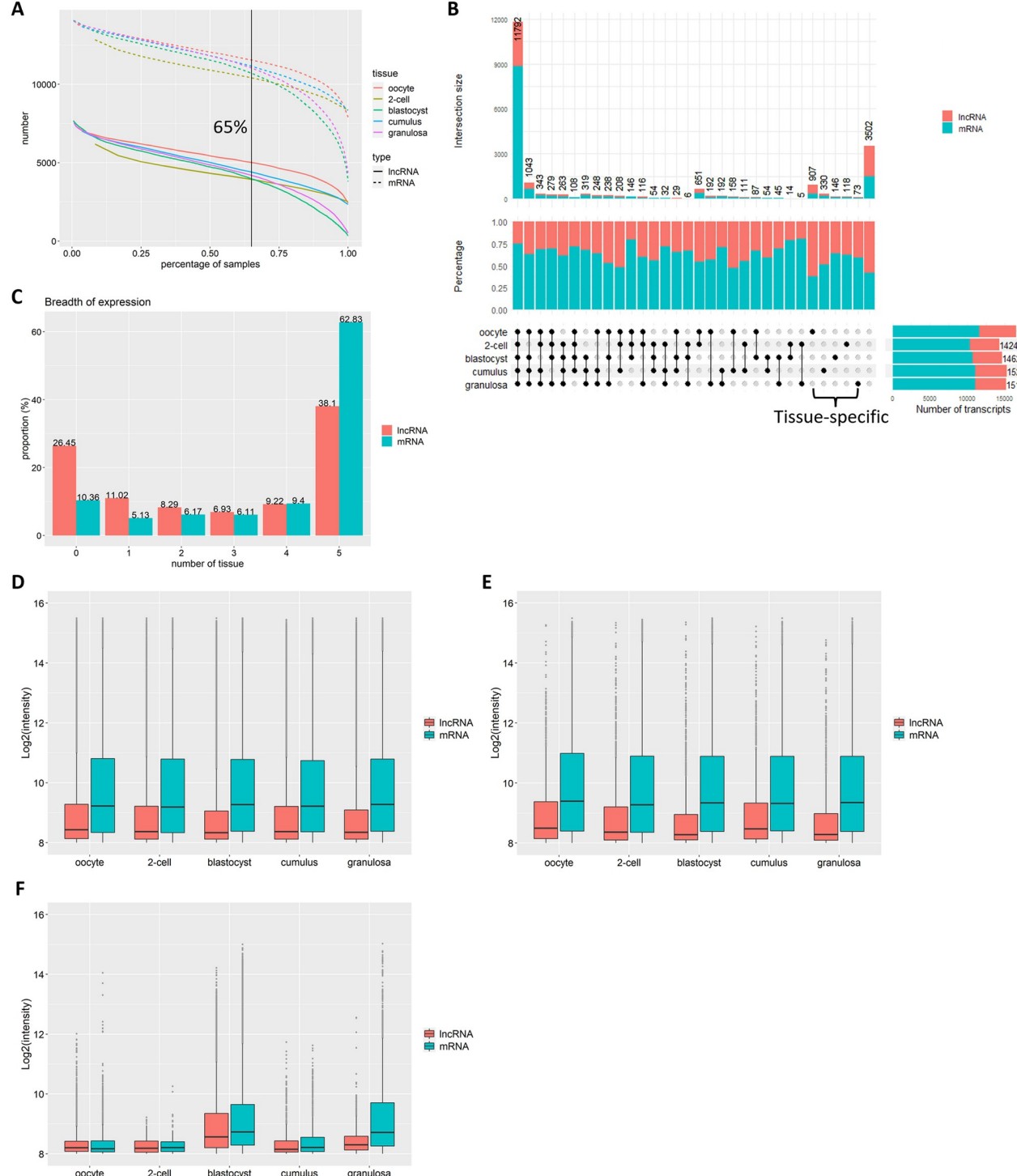

**Fig 4. Abundance of mRNAs and lncRNAs in bovine embryonic cells and tissues.** (A) Number of transcripts expressed in each cell or tissue type versus the proportion of samples expressing the transcripts. (B) ComplexUpset plot of transcript numbers and percentages categorized according to expression status (yes or no) in each cell or tissue type. (C) Frequency of lncRNAs and mRNAs expression categorized according to the number of cell or tissue types for which at least 65% of samples gave a positive signal. (D) Overall expression intensity of mRNAs and lncRNAs expressed in each cell or tissue type. (E) Expression intensity of mRNAs and lncRNAs expressed in all five tissues (n = 11,792). (F) Expression intensity of the tissue-specific transcripts.

other module (S2C Fig). The number of transcripts per module varied widely from 3,514 (2,151 mRNAs, 1,149 lncRNAs, and 214 transcription factors, turquoise) to 49 (43 mRNAs, 2 lncRNAs, and 4 transcription factors, dark grey). The number of identified transcription factors totaled 985, which is 70% of the number in the bovine genome. Ranking the modules by their respective proportions of lncRNAs (Fig 5A) revealed substantial differences. In most modules, mRNAs were found in higher numbers than lncRNAs, as expected. The blue module contains the largest contingent of lncRNAs, accounting for about 56% of all transcripts whereas the proportion of lncRNAs was only 3% in the grey60 module. Moreover, modules showed unequal expression levels and unique expression patterns across tissues (Fig 5B, S3 Fig).

The relationship between tissue specificity and co-expression modules is apparent in Fig 5C. Turquoise, dark green, green, and brown modules were the most significant, associated respectively with oocytes, 2-cell embryos, blastocysts, and granulosa cells based on correlation coefficients. Although lncRNAs were more tissue-specific, the modules containing the most lncRNAs (on the left in Fig 5A–5D) were not those most strongly associated with tissue specificity. In fact, the tissue-related modules are in the middle of Fig 5A–5D and contain a lower proportion of lncRNAs (< 40%, Fig 5C).

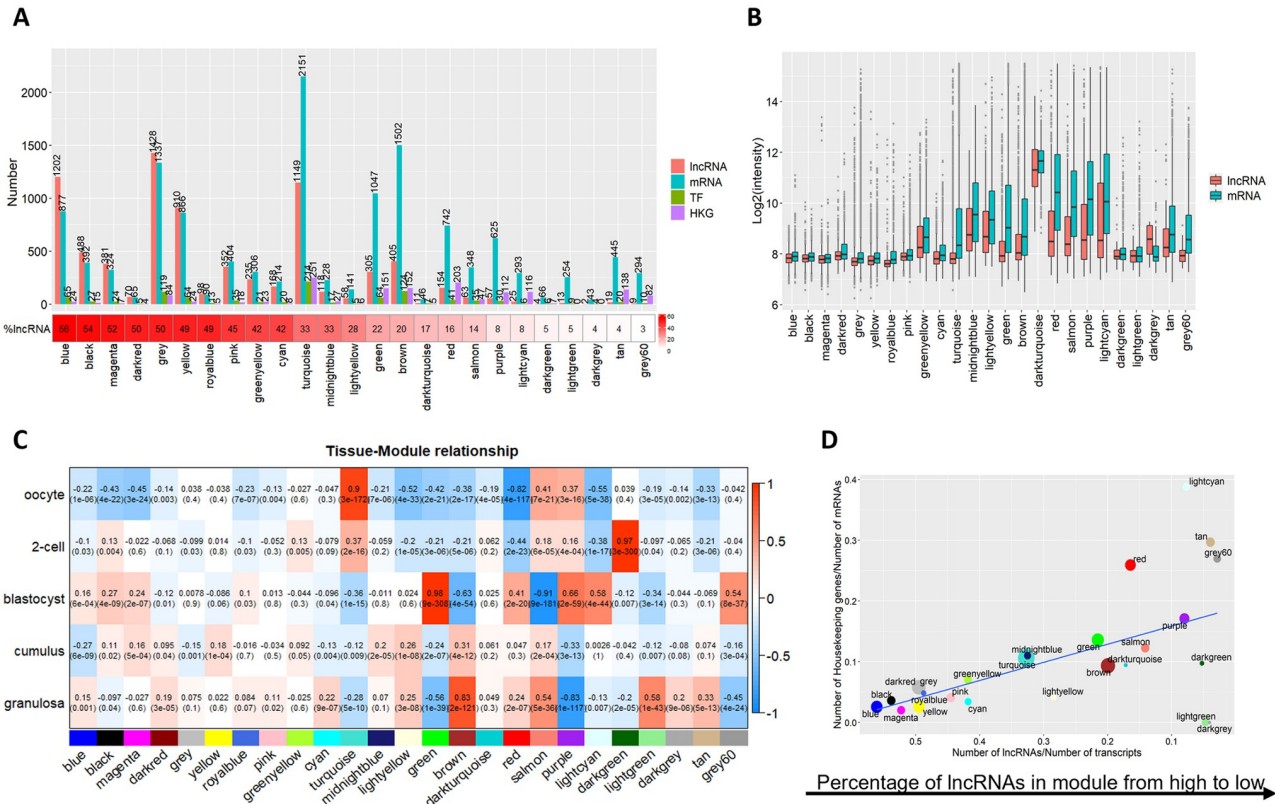

**Fig 5. Co-expression of mRNAs, lncRNAs, transcription factors (TF) and housekeeping genes (HKG) as determined using weighted gene co-expression network analysis.** The expression profiles of the 14,085 mRNAs and 7,724 lncRNAs detected by the microarray are grouped into 25 modules. (A) Module size (number of transcripts) and the percentage of lncRNAs. (B) RNA expression level. (C) Heatmap of co-expression in each cell or tissue type. Red is perfect positive correlation and blue is perfect negative correlation. Correlation coefficients and p (in parentheses) are given. (D) The fraction of housekeeping genes versus the fraction of lncRNAs. The area of the dot is proportional to the number of transcripts in the module.

We then investigated whether lncRNA-riched modules that were not strongly associated with any tissues could be associated with common general cellular functions. Using the HRT Atlas of housekeeping genes and reference transcripts [38], 1,509 housekeeping genes were identified in our dataset. Associating these with the different modules, we can see that their proportion in a module is negatively correlated (Pearson's ρ = -0.57, $P$ = 0.003) with the proportion of lncRNAs (Fig 5D), which corroborates previous research showing that lncRNAs have fewer functions associated with housekeeping genes and have more specific functions [39].

## Functional enrichment analysis

Gene ontology and KEGG enrichment analyses focused on the protein-coding genes in each module were performed to infer indirectly the functions of co-expressed lncRNAs. Based on this analysis, 78 biological processes, 53 cellular components, and 27 molecular functions obtained from gene ontology as well as 61 KEGG pathways were significantly enriched within at least one of the 24 modules (adjusted p-value < 0.05, q-value < 0.2, Fig 6A–6E, S1 Dataset). The first 10 lncRNA-enriched modules reported less enrichment than the others (Fig 6A), which might be due in part to their lower proportions of mRNAs, although their numbers of mRNAs were nevertheless considerable (Fig 5A). Interestingly, mRNAs in modules with a high proportion of lncRNAs tend to be associated with fewer GO and KEGG annotations (S4 Fig). The lesser enrichment in those modules could be due to lncRNAs tending to be co-expressed with mRNAs that are less known or involved in a broad spectrum of different functions. The modules on the left are associated with visual perception, olfactory and taste transduction, axon, glutamatergic synapse, and neuroactive ligand-receptor interaction, suggesting neuronal functions. This apparent relationship between ovarian physiology and neuronal function has been reported previously [40, 41].

Where lncRNAs are less abundant, as in the turquoise module (associated with the oocyte, Fig 5C), enrichments are related to meiosis, protein kinase and ubiquitin-dependent turnover, RNA helicase and DNA binding/RNA transcription, and beta-catenin binding. The light cyan module (associated with blastocysts) is enriched for mitochondrial respiration and initiation of translation functions, elements known to be key features of pre-hatching development following activation of the embryonic genome [42]. The light green and brown modules (associated with granulosa cells) are enriched for immune-related functions and transforming growth factor signaling. We have shown here that most enrichments that are associated to the different modules are relevant for ovarian follicles and early embryos development and could be associated as putative functions to the lncRNAs.

## Construction of lncRNAs-mRNAs-TFs co-expression network and detection of hub transcription factors and hub genes

Another way to infer potential functions for lncRNAs is to test networks to identify hub genes, that is, genes that are functionally interconnected with numerous other genes and therefore important because of their centrality within a network of genes. We calculated the Pearson correlation coefficient of each pair of transcripts in each module to reveal interactions between lncRNAs, mRNAs and transcription factors, defined as significant if ρ > 0.7 and p-value < 0.05. The numbers of significant pairings are shown in Table 1. This also allowed us to identify the transcription factor connected to the largest number of lncRNAs (lncRNA-TF) and mRNAs (mRNA-TF) as well as the most connected lncRNA and mRNA. The hub transcription factors and hub genes found in the present study will be discussed in more detail below.

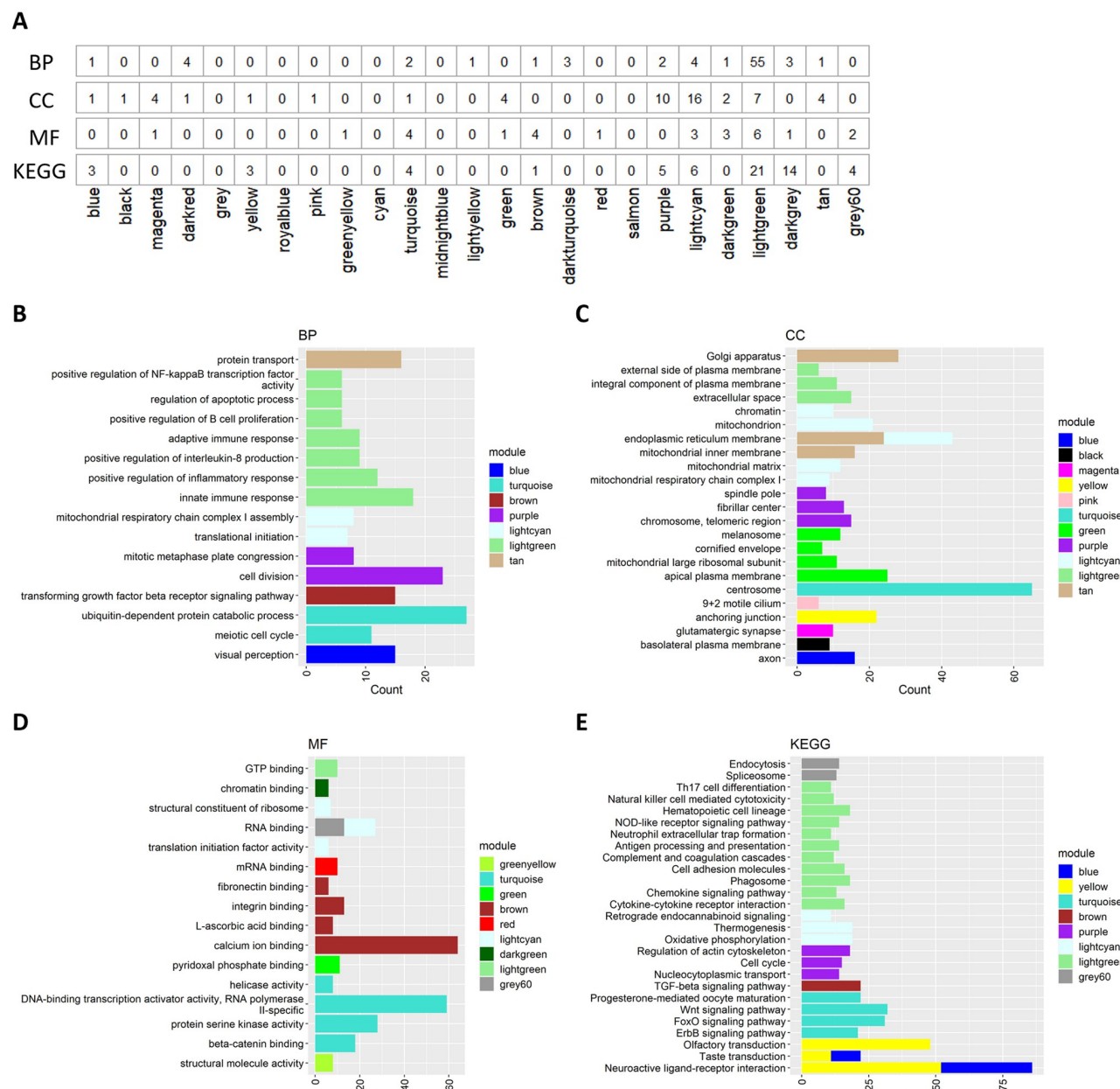

**Fig 6. Summary of the gene ontology and Kyoto Encyclopedia of Genes and Genomes (KEGG) enrichment results.** (A) Number of biological processes (BP), cellular component (CC), molecular function (MF) and KEGG pathway gene expressions enriched in each module. (B-D) Gene Ontology enrichment results. (E) KEGG enrichment results.

## Discussion

In this study, we built a bioinformatic workflow to reannotate all probes on the EmbryoGENE microarray, which contains a comprehensive probe set designed to query a large contingent of long non-coding RNAs [28]. All probes were categorized as protein-coding genes or lncRNAs. This integration process is generally very challenging due to batch effects, that is, technical sources of variation due to differences in sample preparation protocol and data generation on different technological platforms [43, 44]. The advantage of our dataset is that all samples were

**Table 1. WGCNA modules, number of transcripts with significant correlations, hub transcription factors and hub genes.**

| Module | lncRNA | mRNA | TF | lncRNA hub TF | mRNA hub TF | Hub lncRNA | Hub mRNA |
|---|---|---|---|---|---|---|---|
| Blue | 418 | 311 | 23 | NEUROD6 | GMEB2 | Chr4: 86594210-86594269 | *CXHXorf36* |
| Black | 295 | 213 | 13 | MAF | PDX1 | Chr20: 40288655-40288714 | *ENSBTAG00000049694* |
| Magenta | 233 | 190 | 15 | ASCL4 | ASCL4 | Chr22: 59026297-59026356 | *NANOS3* |
| Dark red | 68 | 63 | 5 | HOXA13 | HOXA13 | Chr24: 39937097-39937152 | *SPG21* |
| Yellow | 551 | 490 | 41 | MYOG | MYOG | Chr10: 100009044-100009104 | *GRID1* |
| Royal blue | 12 | 27 | 2 | EPAS1 | MTA2 | ChrX: 4920132-4920191 | *RSPH6A* |
| Pink | 241 | 272 | 26 | ZZZ3 | NPAS4 | Chr23: 13700972-13701031 | *CDH9* |
| Green yellow | 153 | 212 | 14 | FEZF2 | FEZF2 | Chr19: 19038074-19038133 | *JPH4* |
| Cyan | 133 | 168 | 18 | RAX2 | RAX2 | Chr29: 46368869-46368927 | *GPHB5* |
| Turquoise | 824 | 1735 | 163 | FIGLA | ALX1 | Chr2: 85397100-85397159 | *WEE2* |
| Midnight blue | 106 | 216 | 13 | HOXC4 | HOXC4 | ChrX: 34600583-34600641 | *XDH* |
| Light yellow | 42 | 115 | 4 | MYCL | MYCL | Chr4: 68424692-68424746 | *RC3H1* |
| Green | 138 | 758 | 45 | TFAP2C | TFAP2C | Chr7: 13567179-13567238 | *TFAP2C* |
| Brown | 222 | 1112 | 85 | ZBTB41 | TOX2 | Chr21: 8086494-8086553 | *TUSC3* |
| Dark turquoise | 11 | 46 | 7 | TWIST2 | ZBTB16 | Chr10: 79555257-79555316 | *EPHB4* |
| Red | 62 | 564 | 30 | AEBP2 | GATA6 | Chr5: 62754522-62754581 | *DESI2* |
| Salmon | 42 | 293 | 27 | PHTF1 | PHTF1 | Chr18: 47416784-47416843 | *LIX1L* |
| Purple | 31 | 491 | 17 | GATA2 | GATA2 | Chr1: 85301605-85301664 | *FAM169A* |

*(Continued)*

**Table 1.** (Continued)

| Module | lncRNA | mRNA | TF | lncRNA hub TF | mRNA hub TF | Hub lncRNA | Hub mRNA |
|---|---|---|---|---|---|---|---|
| Light cyan | 14 | 242 | 3 | ATF4 | TP53 | Chr4: 77120593-77120652 | *EIF3D* |
| Dark green | 4 | 43 | 5 | NR6A1 | AFF1 | Chr9: 93953194-93953253 | *IGF2BP2* |
| Light green | 11 | 208 | 8 | IKZF1 | SPI1 | Chr5: 75655064-75655123 | *LAPTM5* |
| Dark grey | 2 | 32 | 1 | ATOH1 | ATOH1 | Chr2: 119182807-119182866 | *UGT2B10* |
| Tan | 8 | 345 | 16 | MZF1 | RBCK1 | Chr8: 74841676-74841735 | *NENF* |
| Grey60 | 3 | 221 | 7 | MSX2 | MSX2 | Chr19: 27370446-27370505 | *SRM* |

processed using the same protocols and technological platform, making datasets more homogeneous and compatible and less confounded when merged.

After reannotating, assembling and curating all 468 datasets, 21,809 probes were found to be expressed, of which 7,724 are lncRNAs, making the EmbryoGENE microarray very suitable for studying these uncharacterized elements. Reannotations of microarrays, not originally designed for lncRNA studies, have proven valuable in uncovering useful information on these transcripts. For example, reannotating Affymetrix Mouse Genome Array probes led to the prediction of function for 340 lncRNAs [45]. The present study demonstrates that analyzing microarray data from multiple experiments is a powerful method for investigating lncRNAs function.

Using the EmbryoGENE microarray platform to study physiological contrasts has previously revealed lncRNAs among differentially expressed transcripts [46–49] and one unexpected function whereby the reduction of abundance by RNA silencing increased the rate of embryonic development [50]. However, the dataset has yet to be integrated entirely.

As shown previously and by the present results, lncRNAs expression is lower in all tissues compared to mRNAs and tissue-specific lncRNAs are often less abundant than ubiquitously expressed lncRNAs [7, 51]. The finding that lncRNAs could be better than mRNAs as a marker of specific tissue types (Fig 3) was unexpected. Such specificity has been proposed previously [52]. The evidence that brain and testis express more tissue-specific lncRNAs [53, 54] suggests that such transcripts might be important for the acquisition of specific phenotypic traits. Altered lncRNAs expression observed in numerous diseases [55] and the high tissue specificity suggest important regulatory roles rather than non-specific artefacts of leaky transcription.

Physiological functions of lncRNAs may be inferred by examining co-expressed mRNAs under the assumption that highly co-expressed transcripts are more likely to be functionally associated [56]. The module with the highest number of co-expressed transcripts (turquoise) is highly correlated with the oocyte stage. The enrichment of gene ontology terms suggests the participation of these lncRNAs in meiosis, ubiquitin-dependent protein catabolism,

centrosome, helicase activity and DNA-binding transcription activator activity (RNA polymerase II-specific), which are all related to DNA remodeling events. These are essential for oogenesis, during which transcriptional activity is increased until later stages of folliculogenesis (e.g., late antral phase) when oocyte DNA condenses, transcription ceases, and meiosis is underway leading to the formation of the metaphase II oocyte awaiting fertilization [57, 58]. The ubiquitin-proteasome pathway reportedly can control oocyte meiotic maturation, oocyte-sperm binding, and early embryo development [59, 60]. Several key pathway enrichments shown in Fig 6E are related to oocyte maturation, including "FoxO signaling" [61], "Wnt signaling" [62] and "progesterone-mediated oocyte maturation". Progesterone triggers oocyte maturation in *Xenopus*. In mammals, the LH surge (Fig 6E) does this [63]. All core cellular processes known to be key for the production of the female gamete are highlighted (turquoise module, S2 Dataset), namely the known oocyte-specific genes *PHACTR3*, *PTPRQ*, *WEE2*, *KPNA7*, *DAZL*, *MOS*, *ZAR1*, *FIGLA*, *GDF9*, *BMP15*, the zona pellucida genes *ZP3* and *ZP4* [64–69] and aurora kinases (*AURKB*, *AURKC*) which are required for oocyte meiotic maturation [70] and the NLR family (*NLRP2*, *5*, *7*, *8*, *9*, and *13*) which are important in oogenesis and early embryo development [71–73].

FIGLA, identified as the hub transcription factor associated with the largest number of lncRNA transcripts in the turquoise module (lncRNA-TF in Table 1), is folliculogenesis specific, expressed preferentially in oocytes and regulates multiple oocyte-specific genes. Expression of *Sirena1*, the most abundant lncRNA in mouse oocytes, depends on the FIGLA, LHX8, and NOBOX maternal transcription factors network [67].

Enrichment in the light cyan module (Fig 6B) indicates possible involvement of lncRNA in translation, mitochondrial respiration, and mitochondrial gene expression. Based on the module-tissue relationship (Fig 5C), the light cyan module was closely related to the blastocyst stage. Overrepresented functional categories such as ribosomes, mitochondria and oxidative phosphorylation have been found prevalent among highly expressed genes in bovine blastocysts, indicating that these cells are rapidly synthesizing proteins to sustain high rates of growth and division [74]. Likewise, the hub gene in the light cyan module is *EIF3D* (Table 1), which is required for initiating translation [75]. Similarly, enrichment analyses pointed to involvement of the light green module lncRNAs in the immune system. The hub transcription factors linking the most lncRNAs and mRNAs in this module were respectively IKZF1 and SPI1 (Table 1), suggesting that both types of transcripts could be regulated by different factors. Also found in this module, the hub gene *LAPTM5* is expressed preferentially in immune cells [76]. Based on the module-tissue relationship, the light green module is closely related to granulosa cells, corroborating the finding that granulosa cells can identify danger signals and undertake immune cell functions [77]. These results indicate that lncRNAs could be involved in these essential steps along with key proteins. Several lncRNAs expressed in cumulus and granulosa cells are reportedly involved in essential pathways that contribute to oocyte maturation and embryo development. For example, *AK124742*, an antisense of PSMD6, was upregulated in high-quality embryos compared to poor-quality embryos [78]. In addition, *NEAT1*, *MALAT1*, *ANXA2P2*, *MEG3*, *IL6STP1*, and *VIM-AS1* might be involved in apoptosis and extracellular matrix-related functions and appear to be essential for oocyte growth [79]. Our results highlight that lncRNAs are co-expressed strongly with key genes that might be involved in folliculogenesis and early embryogenesis (Fig 7). Although regulatory roles of lncRNAs need to be validated experimentally, for example by knockdown or overexpression, computational annotation based on correlated expression remains the most intuitive and commonly used method of imputing function to lncRNAs. This should at least guide investigations into function.

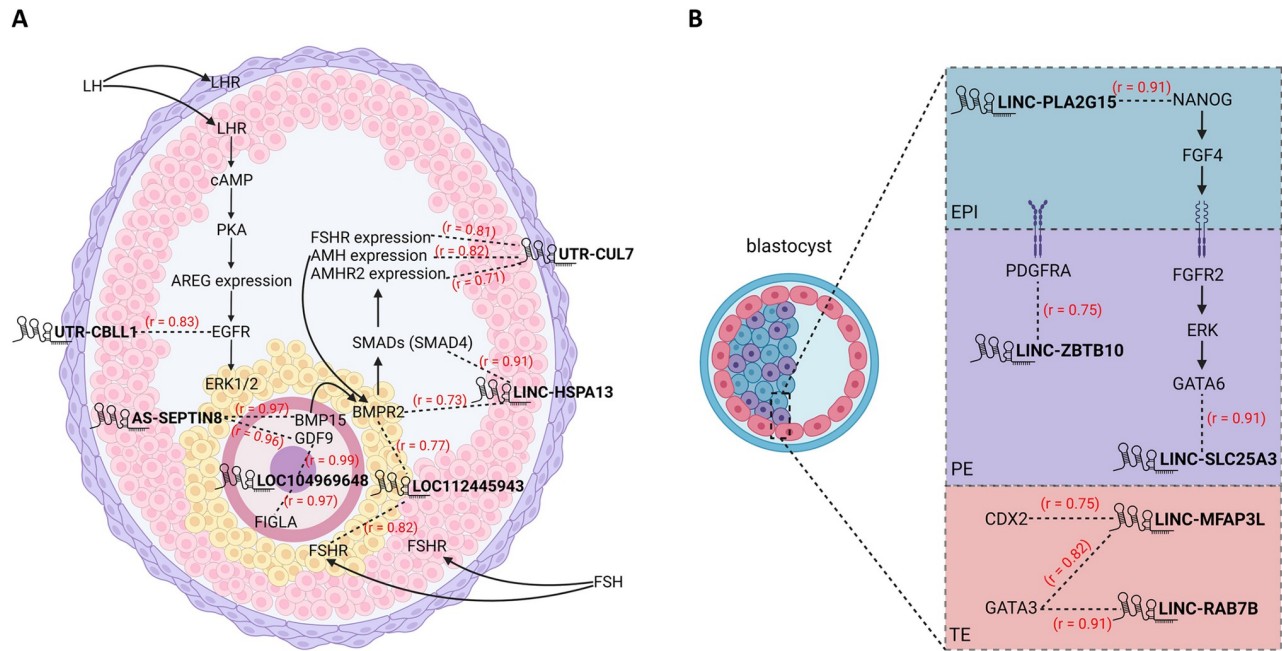

**Fig 7. (A) Examples of potential lncRNAs involvement in folliculogenesis and (B) blastocyst formation.** EPI: pluripotent epiblast; PE: primitive endoderm; TE: trophectoderm. RNA is in bold type. Correlation coefficients are in red.

## Conclusions

It has been proposed that changes in lncRNAs expression are equal in importance to mRNAs for understanding folliculogenesis, oogenesis and early development [46, 80, 81]. The results of the present study globally support this claim and reveal that a large contingent of lncRNAs are present during folliculogenesis, oogenesis and early embryonic development in cattle. The strong correlations with key protein-coding genes known to be essential for reproduction suggest that some lncRNAs must play essential roles. How these transcripts affect their physiological functions are still largely unknown. In the nucleus, several lncRNAs are known to be involved in RNA processing hence exerting some control over gene expression [82] but their role in the cytoplasm is much less understood. It has been proposed that some may act as adaptor molecules in protein complexes alongside RNA binding proteins [83]. Given that different classes of lncRNAs exist (e.g., lincRNA, intron derived, antisense, etc.), they should be expected to be involved in various cellular functions.

## Supporting information

**S1 Fig.** (A) Correlation analysis across datasets (N = 62). (B) Correlation analysis across samples (N = 473). The arrow represents outlier samples.
(TIF)

**S2 Fig.** (A) Scale-free topology model fit for determining the optimal soft threshold. (B) mean connectivity analysis for selecting the optimal soft threshold. (C) The cluster dendrogram and module colors of 21809 transcripts.
(TIF)

**S3 Fig. Expression patterns of lncRNAs and mRNAs across tissues in modules.** The lines represent the mean expression levels. Modules are arranged by decreasing fraction of

lncRNAs. The bars above each graph show the proportion of lncRNAs and mRNAs in each module.
(TIF)

**S4 Fig.** (A) the fraction of mRNAs with BP annotations. (B) the fraction of mRNAs with CC annotations. (C) the fraction of mRNAs with MF annotations. (D) the fraction of mRNAs with KEGG annotations. The blue line represents the linear regression. Abbreviations: BP, biological process; CC, cellular component; MF, molecular function; KEGG, Kyoto Encyclopedia of Genes and Genomes.
(TIF)

**S1 Table. Definition and number of probes for different types.**
(DOCX)

**S1 Dataset. The complete functional enrichment results in 24 modules.**
(XLSX)

**S2 Dataset. Probe information.**
(XLSX)

## Acknowledgments

We thank all members of the EmbryoGENE project for giving public access to all the data used in this publication.

## Author Contributions

**Conceptualization:** Pengmin Wang, Éric R. Paquet, Claude Robert.

**Data curation:** Pengmin Wang, Éric R. Paquet.

**Formal analysis:** Pengmin Wang, Éric R. Paquet, Claude Robert.

**Funding acquisition:** Claude Robert.

**Investigation:** Pengmin Wang, Éric R. Paquet, Claude Robert.

**Methodology:** Pengmin Wang, Éric R. Paquet, Claude Robert.

**Project administration:** Claude Robert.

**Resources:** Pengmin Wang, Éric R. Paquet, Claude Robert.

**Software:** Pengmin Wang, Éric R. Paquet.

**Supervision:** Éric R. Paquet, Claude Robert.

**Validation:** Pengmin Wang, Éric R. Paquet, Claude Robert.

**Visualization:** Pengmin Wang, Éric R. Paquet.

**Writing – original draft:** Pengmin Wang, Éric R. Paquet, Claude Robert.

**Writing – review & editing:** Pengmin Wang, Éric R. Paquet, Claude Robert.

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
