## [Decision Letter · Decision Letter 0]

17 Jul 2023

PONE-D-23-14575May 17, 2023 PLOS One

Comprehensive transcriptomic analysis of long non-coding RNAs in bovine ovarian follicles and early embryosPLOS ONE

Dear Dr. Robert,

Thank you for submitting your manuscript to PLOS ONE. After careful consideration, we feel that it has merit but does not fully meet PLOS ONE’s publication criteria as it currently stands. Therefore, we invite you to submit a revised version of the manuscript that addresses the points raised during the review process.

We look forward to receiving your revised manuscript.

Kind regards,

Xiaoyong Sun

Academic Editor

PLOS ONE

2. Please amend either the title on the online submission form (via Edit Submission) or the title in the manuscript so that they are identical.

Reviewers' comments:

Reviewer's Responses to Questions

**Comments to the Author**

1. Is the manuscript technically sound, and do the data support the conclusions?

Reviewer #1: Yes

Reviewer #2: Yes

Reviewer #3: Partly

2. Has the statistical analysis been performed appropriately and rigorously? 

Reviewer #1: Yes

Reviewer #2: Yes

Reviewer #3: No

3. Have the authors made all data underlying the findings in their manuscript fully available?

Reviewer #1: Yes

Reviewer #2: Yes

Reviewer #3: Yes

4. Is the manuscript presented in an intelligible fashion and written in standard English?

Reviewer #1: Yes

Reviewer #2: Yes

Reviewer #3: Yes

5. Review Comments to the Author

Reviewer #1: Dear Editor,

In the study entitled "Comprehensive transcriptomic analysis of long non-coding RNAs in bovine ovarian follicles and early embryos," the authors have briefly described and found out about comprehensive analysis and reviewed lncRNA expression in bovine ovarian follicles and early embryos. The analysis revealed that protein-coding genes and lncRNAs are both expressed more in oocytes. Additionally, this provides evidence that lncRNAs play diverse physiological roles that are tissue-specific and associated with key cellular functions alongside mRNAs in bovine ovarian follicles and early embryos. The overall manuscript is well written; however, many minor arguments in the article should be improved to make the manuscript more expressive. Authors need to do English language checking or improve the English language with the help of native speakers because there are few spelling and typographical mistakes in the manuscript.

Abbreviations should be written as full terms (abbreviations) when used for the first time in the text.

i) The scientific name should be italicized in the whole manuscript; check carefully.

ii) Keywords should preferably be different from the main title.

iii) The word “non-coding” or “noncoding”, pattern must be the same throughout the manuscript.

On lines 27–28, the authors need to rewrite the sentence because few words are replicated in one sentence.

On line 48, there is a bullet. What does that mean? Same bullet in line (359).

Line 52 authors mentioned (The more we study lncRNAs), but I think it is a previous study, not one we study.

Line 56: The indefinite article “a” may not be required with the plural noun half-lives.

On lines 62–67, there are two big sentences without references. Authors need to add references.

Line 75: It appears that you are missing a comma or two with the interrupter therefore. Consider adding the comma.

Line 93. It seems that you are missing a comma. Consider adding a comma.

Lines 156–162 are not results; authors must describe results rather than material and methods.

Line 158. The to-infinitive to study has been split by the modifier specifically. Avoiding split infinitives can help your writing sound more formal.

Line 162. The phrase researchers of this consortium may be wordy. Consider changing the wording with only “consortium researchers”

Line 174. It appears that p value is missing a hyphen.

Line 186. Please add reference for the statement.

Line 191. It may be unclear to the reader what readily is modifying. Consider moving the modifier.

Line (209-211) There is no need to write here because the authors already explained it in M&M.

Line 214: It seems that the preposition used may be incorrect here. Add “storing of a large amount…”

Line 251. The verb reveals does not agree with the subject. Consider changing the verb form.

Line 263. It seems that you are missing a verb. Consider adding it. “modules were not strongly associated”

Line 302. Add “are” before associated.

Line 336. Remove “That” before were.

Line 357. The phrase Gene ontology terms enrichment points appears to be a confusing noun string. Consider rewriting the sentence for clarity.

Line 360. It appears that clearly may be unnecessary in this sentence. Consider removing it.

Line (370 to 374) Genes names should be in italics.

Line 417. Change “is” in to “are still”.

Line 343. The word the doesn’t seem to fit this context

Reviewer #2: Your article is comprehensive and provides a detailed analysis of long non-coding RNAs (lncRNAs) in bovine ovarian follicles and early embryos. Here are some questions and suggestions that could help you further improve your article:

1. Could you provide a brief overview in abstract of the significance and potential applications of studying lncRNAs in the context of bovine ovarian follicles and early embryos? This would help readers understand the importance of your research.

2. Can you provide a more explicit research gap or knowledge deficiency that your study aims to address? This will help readers understand the unique contribution of your research.

3. It would be helpful to include a sentence or two summarizing the objective of your study. What specific aspects of lncRNA expression and function in bovine ovarian follicles and early embryos are you investigating?

4. Could you provide more details about the sources of the RNA samples used in your study? For example, how were the samples obtained, and were there any specific criteria for their selection?

5. In the reannotation and curation process of the microarray probes, how were the probes classified as mRNAs or lncRNAs? Were there any specific criteria for this classification?

6. Can you explain why a soft thresholding power of 12 was chosen for constructing the co-expression network using WGCNA? Were any sensitivity analyses performed to evaluate the robustness of the network?

7. In the functional enrichment analysis, can you provide more information on the statistical methods used for assessing the significance of enrichment?

8. It would be helpful to summarize the main findings of your study in a concise manner at the beginning of the Results section. This will allow readers to quickly grasp the key outcomes of your research.

9. When discussing the functional enrichment analysis results, can you provide some specific examples of enriched biological processes or pathways that are relevant to bovine ovarian follicles and early embryos? This will help readers understand the functional implications of the identified lncRNAs.

Reviewer #3: This study presents a transcriptomic analysis of coding and long non-coding RNAs (lncRNAs) in bovine ovarian follicles and early embryos. The authors reannotated the probes and evaluated the expression and co-expression of lncRNAs based on a unique database comprising 468 microarray hybridizations. However, there are several concerns that limit the overall quality and relevance of the study, which are outlined below:

1. The paper lacks overall novelty, and it is not clear what the main contribution of this study is to the field. The biological relevance of the results should be further explored to establish their significance.

2. The authors reannotated the probes of the EmbryoGENE microarray platform based on the new version of the genome. While this strategy has the potential to provide more accurate findings, the authors failed to compare their new annotation with the previous annotation provided by the manufacturer of the microarray. Additionally, they need to demonstrate the superiority of their annotation.

3. The study integrated 468 transcriptomes from 47 different experimental conditions. Although all samples were processed using the same protocols and technological platform, batch effects still need to be addressed to ensure accurate results. The authors should provide details of their normalization strategy to mitigate these batch effects.

4. In Figure 3, the authors need to provide more information to facilitate the comparison of different results. For example, in Figure 3A, the contribution of each group of transcripts to PC1 and PC2 is unclear. This information would be helpful in comparing Figure 3A and 3B, as it would reveal which transcripts cluster the different developmental states.

5. The methods section should include an explanation of how the authors performed the correlation of Weighted Gene Co-expression Network Analysis (WGCNA) modules and the groups of samples (categorical variables).

6. Finally, the quality of the figures needs improvement to enhance their readability.

6. PLOS authors have the option to publish the peer review history of their article (what does this mean?). If published, this will include your full peer review and any attached files.

Reviewer #1: **Yes: **Hongwei Zhao

Reviewer #2: No

Reviewer #3: No

---

## [Author Response · Author response to Decision Letter 0]

28 Aug 2023

Dear Editor, 

Please find below our answers to the reviewers’s. We found that everything was relevant and positive. This new version is improved. 

Sincerely, 

Claude Robert

Reviewer #1

i) The scientific name should be italicized in the whole manuscript; check carefully.

Authors: The in vitro on line 69 was italicized as well as all mRNA and lncRNA gene names (eg. in Table 1). This is in line with the scientific gene nomenclature convention: https://en.wikipedia.org/wiki/Gene_nomenclature

ii) Keywords should preferably be different from the main title.

Authors: Thanks. We changed the keywords to: Long non-coding RNA, gene expression, ovarian follicle, oocyte, blastocyst.

iii) The word “non-coding” or “noncoding”, pattern must be the same throughout the manuscript.

Authors: Thanks for pointing this out. We changed noncoding to “non-coding” throughout the entire manuscript. 

On lines 27–28, the authors need to rewrite the sentence because few words are replicated in one sentence.

Authors: The sentence was rewritten (lines 28-29).

On line 48, there is a bullet. What does that mean? Same bullet in line (359).

Authors: These are the symbol Ⅱ, the roman numeral two and it is ok in Word, but not in the pdf sent to the reviewer. We changed it to two capital “i” instead and it looks ok now (line 51 and line 376).

Line 52 authors mentioned (The more we study lncRNAs), but I think it is a previous study, not one we study.

Authors: You are correct. This statement is based on the growing literature on lncRNAs. The phrase was modified (line 55). 

Line 56: The indefinite article “a” may not be required with the plural noun half-lives.

Authors: The “a” was deleted (line 54). 

On lines 62–67, there are two big sentences without references. Authors need to add references.

Authors: Reference #26 was added.

Line 75: It appears that you are missing a comma or two with the interrupter therefore. Consider adding the comma.

Authors: A comma was added on line 82.

Line 93. It seems that you are missing a comma. Consider adding a comma.

Authors: The modification was made (line 100). 

Lines 156–162 are not results; authors must describe results rather than material and methods.

Authors: We understand the reviewer’s point. We tested removing the phrases, but it always comes back to the point the platform is not widely known. To fully appreciate the result, we believe it is important to briefly describe what is this platform exactly. Similarly, when a novel method is used, it is not uncommon to provide a brief description of what it does. However, we acknowledge the phrases could still be removed and the work will still be understandable. Some readers could need to go back to the Material and Methods to know more. This said, if the reviewer feels it is unacceptable to describe the platform that was used to generate the database, let us know and we will comply and remove it. At this point, we would prefer keeping it (lines 168-175).

Line 158. The to-infinitive to study has been split by the modifier specifically. Avoiding split infinitives can help your writing sound more formal.

Authors: We removed “specifically” from this sentence (line 171).

Line 162. The phrase researchers of this consortium may be wordy. Consider changing the wording with only “consortium researchers”

Authors: The proposed modification was made (line 175). 

Line 174. It appears that p value is missing a hyphen.

Authors: The proposed modification was made (line 187). 

Line 186. Please add reference for the statement.

Authors: Reference #35 was added. 

Line 191. It may be unclear to the reader what readily is modifying. Consider moving the modifier.

Authors: The sentence was modified (line 204).

Line (209-211) There is no need to write here because the authors already explained it in M&M.

Authors: The sentence was deleted (lines 223-225).

Line 214: It seems that the preposition used may be incorrect here. Add “storing of a large amount…”

Authors: The proposed modification was made (line 228). 

Line 251. The verb reveals does not agree with the subject. Consider changing the verb form.

Authors: The verb form was changed to past tense (line 265). 

Line 263. It seems that you are missing a verb. Consider adding it. “modules were not strongly associated”

Authors: The sentence was modified (line 277).

Line 302. Add “are” before associated.

Authors: “which are” has been added (line 317).

Line 336. Remove “That” before were.

Authors: The sentence was modified (lines 351-353).

Line 357. The phrase Gene ontology terms enrichment points appears to be a confusing noun string. Consider rewriting the sentence for clarity.

Authors: The sentence was modified (lines 373-374).

Line 360. It appears that clearly may be unnecessary in this sentence. Consider removing it.

Authors: The modification was made (line 377). 

Line (370 to 374) Genes names should be in italics.

Authors: Gene names have been italicized (lines 387-390). 

Line 417. Change “is” in to “are still”.

Authors: The modification was made (line 434).

Line 343. The word the doesn’t seem to fit this context

Authors: “Using the EmbryoGENE microarrays…” was modified to “Using the EmbryoGENE microarray platform…” (line 358). 

Reviewer #2

1. Could you provide a brief overview in abstract of the significance and potential applications of studying lncRNAs in the context of bovine ovarian follicles and early embryos? This would help readers understand the importance of your research.

Authors: We added a sentence at the end of the abstract (lines 34-36). 

2. Can you provide a more explicit research gap or knowledge deficiency that your study aims to address? This will help readers understand the unique contribution of your research.

Authors: We fully agree this was missing. A statement on the value of the work has been added (lines 76-81).

3. It would be helpful to include a sentence or two summarizing the objective of your study. What specific aspects of lncRNA expression and function in bovine ovarian follicles and early embryos are you investigating?

Authors: Precisions on the objectives was added (lines 167-168). 

4. Could you provide more details about the sources of the RNA samples used in your study? For example, how were the samples obtained, and were there any specific criteria for their selection?

Authors: Details about the platform were added on lines 111-114. We are presenting a meta-analysis of previous physiological contrasts done by numerous investigators over several years. All experiments were done using the same platform using the same sample processing procedures. These methods were derived from tests done when developing the platform This work can be found in the following publications: PMID: 20479066 and PMID: 21812063). 

In total, we accounted for 468 transcriptomes from 47 different experimental conditions and different labs. All samples were bovine ovarian cells and early embryos. We did not collect and process any new samples for this study. Details about the sources of the RNA samples used in each dataset can be found in the respective publications from the listed GEO samples.

5. In the reannotation and curation process of the microarray probes, how were the probes classified as mRNAs or lncRNAs? Were there any specific criteria for this classification?

Authors: This is specified in the Materials and methods between lines 96 and 107. Here is the part of text specifying this information “Using bovine genome annotations obtained from UCSC, Ensembl, and NCBI, probes were then reannotated separately, as mRNAs if they overlapped with at least one exon on the same strand, as intronic lncRNAs if they were within introns of protein-coding genes on the same strand, as antisense lncRNAs if they overlapped with protein-coding genes on the opposite strand, and finally, as intergenic lncRNAs if they did not overlap with protein-coding genes. Probes annotated ambiguously as mRNAs or lncRNAs in the same database were discarded. To lower the risk of errors in identifying lncRNAs, we kept 30,575 probes left after this annotation procedure and assigned them to lncRNAs if all three databases concurred. Only one database was required to assign a probe to mRNA. To reduce redundancy when a plurality of probes targeted the same gene, the probe with the highest mean signal was selected, which reduced the total to 21,840. The entire workflow leading to the final annotation is represented in Fig 1.” We believe this is precise enough and that we are using a well-accepted way to classify probes.

6. Can you explain why a soft thresholding power of 12 was chosen for constructing the co-expression network using WGCNA? Were any sensitivity analyses performed to evaluate the robustness of the network?

Authors: We selected the value of 12 based on a scale-free topology of R2 = 0.9 and mean connectivity < 100 as presented in supplementary Figure S2 A and B and mentioned in the text on line 134 to 135. This is what is recommended by the authors of WGCNA.

7. In the functional enrichment analysis, can you provide more information on the statistical methods used for assessing the significance of enrichment?

Authors: The enricher function in the clusterProfiler package in R, that we use in the paper, uses a classic hypergeometric test to assess significance. We used Benjamini-Hochberg adjusted p-values < 0.05 to select for significant enrichments. We added more information on lines 153-154.

8. It would be helpful to summarize the main findings of your study in a concise manner at the beginning of the Results section. This will allow readers to quickly grasp the key outcomes of your research.

Authors: Our group is not used to summarizing results at the beginning of the results section, but more at the beginning of the discussion or in the abstract. Nevertheless, to comply with this comment, we added one sentence at the beginning of the results section. (lines 167-168). 

9. When discussing the functional enrichment analysis results, can you provide some specific examples of enriched biological processes or pathways that are relevant to bovine ovarian follicles and early embryos? This will help readers understand the functional implications of the identified lncRNAs.

Authors: Most enrichments shown in Figure 6 are relevant to bovine ovarian follicles and early embryos. We believe the requested info for specific examples can be found in the Discussion section. It is noteworthy to mention that we are always interested in what we found rather than what we did not find. But sometimes, a negative result is also intriguing and meaningful. On this angle, is it interesting to note that the clusters containing the most lncRNAs were also populated with lesser-known mRNAs. 

Reviewer #3

1. The paper lacks overall novelty, and it is not clear what the main contribution of this study is to the field. The biological relevance of the results should be further explored to establish their significance.

Authors: We think we understand the comment from this reviewer. Our study lacks experimental validation of the functions of lncRNAs. This is unfortunately common when combining datasets to perform a comprehensive re-analysis. Here we understand that lncRNAs are guilty by association with mRNAs. This is a common approach to study novel uncharacterized features. Here, what can be appreciated is the wealth of the database which provides more power than individual contrasts. It offered a global perspective of the expression of lncRNAs in bovine ovarian follicles and early embryos. This is the steppingstone to select the most interesting and promising ones. By making it public, others could also use the data to select candidates to explore in future studies. We believe all the findings presented in the paper and this resource are still extremely valuable for the community.

2. The authors reannotated the probes of the EmbryoGENE microarray platform based on the new version of the genome. While this strategy has the potential to provide more accurate findings, the authors failed to compare their new annotation with the previous annotation provided by the manufacturer of the microarray. Additionally, they need to demonstrate the superiority of their annotation.

Authors: The previous annotation provided by the manufacturer of the microarray was Bos_taurus_UMD_3.1/bosTau6 (Nov.2009). From the website (https://www.ncbi.nlm.nih.gov/genome/annotation_euk/Bos_taurus/106/), there is a table that summarizes the changes in gene set between ARS-UCD1.2 (current) and Bos_taurus_UMD_3.1.1/bosTau8 (Jun.2014). Only 18% of genes in the current annotation release perfectly matched in exon boundaries with the previous one and 19% are without a match in the previous release, suggesting that the up-to-date annotation is clearly superior. About a compassion with the previous annotation provided by the manufacturer of the microarray, we added a column in the supplementary S2 Dataset and among 14,085 coding genes in our re-analysis, 9,165 of them have the same gene symbol as the previous annotation. We believe that this is sufficient to conclude the reannotation was essential for this analysis to be appropriate.

3. The study integrated 468 transcriptomes from 47 different experimental conditions. Although all samples were processed using the same protocols and technological platform, batch effects still need to be addressed to ensure accurate results. The authors should provide details of their normalization strategy to mitigate these batch effects.

Authors: The batch effect is caused by methodological variance that could be introduced at all steps during sample processing. To better control it, the EmbryoGENE platform was tested to define standard operation procedures that have been used for all samples. This work can be found in the following publications: PMID: 20479066 and PMID: 21812063. The EmbryoGENE microarray is a dual-color platform where two samples were compared on the same array and dye-swapped were done to account for labeling and hybridization. We agree that this does not prevent from batch effects, but it is meant to minimize it. 

In the current study, the entire database was submitted to several QC steps to identify outstanding samples (please refer to Figure 1). Not all transcriptomes were kept. After removing some datasets and samples (S1 Fig), samples are clustered by tissues from PCA plot (Fig 3). Our 468 samples are from 47 different datasets. Among these datasets, GSE27530 includes samples from granulosa and blastocyst. From Figure 3B, blastocysts separate well with granulosa, supporting minimal batch effect. 

4. In Figure 3, the authors need to provide more information to facilitate the comparison of different results. For example, in Figure 3A, the contribution of each group of transcripts to PC1 and PC2 is unclear. This information would be helpful in comparing Figure 3A and 3B, as it would reveal which transcripts cluster the different developmental states.

Authors: Thank you. The information was added in the Results section (lines 206-208).

5. The methods section should include an explanation of how the authors performed the correlation of Weighted Gene Co-expression Network Analysis (WGCNA) modules and the groups of samples (categorical variables).

Authors: The requested information was added in the Materials and methods (line 141).

6. Finally, the quality of the figures needs improvement to enhance their readability.

Authors: Thank you for your comment. We tried to make clear figures. If the reviewer has specific points for improvements, we will consider them. Is the comment aimed at figure resolution? The figures in the PDF are indeed low resolution. But the uploaded files are all 2-3 Mb each and all passed the quality control of PACE which is testing the journal's figure requirements. We believe the PDF is in low resolution to reduce file size.

---

## [Decision Letter · Decision Letter 1]

5 Sep 2023

Comprehensive transcriptomic analysis of long non-coding RNAs in bovine ovarian follicles and early embryos

PONE-D-23-14575R1

Dear Dr. Robert,

We’re pleased to inform you that your manuscript has been judged scientifically suitable for publication and will be formally accepted for publication once it meets all outstanding technical requirements.

Kind regards,

Xiaoyong Sun

Academic Editor

PLOS ONE

Additional Editor Comments (optional):

Reviewers' comments:

Reviewer's Responses to Questions

**Comments to the Author**

1. If the authors have adequately addressed your comments raised in a previous round of review and you feel that this manuscript is now acceptable for publication, you may indicate that here to bypass the “Comments to the Author” section, enter your conflict of interest statement in the “Confidential to Editor” section, and submit your "Accept" recommendation.

Reviewer #1: All comments have been addressed

Reviewer #2: All comments have been addressed

2. Is the manuscript technically sound, and do the data support the conclusions?

Reviewer #1: Yes

Reviewer #2: Yes

3. Has the statistical analysis been performed appropriately and rigorously? 

Reviewer #1: Yes

Reviewer #2: Yes

4. Have the authors made all data underlying the findings in their manuscript fully available?

Reviewer #1: Yes

Reviewer #2: Yes

5. Is the manuscript presented in an intelligible fashion and written in standard English?

Reviewer #1: Yes

Reviewer #2: Yes

6. Review Comments to the Author

Reviewer #1: Since the authors have resolved all minor concerns highlighted during the review process, I am satisfied with the current form of the manuscript and am prepared to accept it

Reviewer #2: (No Response)

7. PLOS authors have the option to publish the peer review history of their article (what does this mean?). If published, this will include your full peer review and any attached files.

Reviewer #1: **Yes: **Hongwei Zhao

Reviewer #2: No

---

## [Editor Report · Acceptance letter]

11 Sep 2023

PONE-D-23-14575R1 

Comprehensive transcriptomic analysis of long non-coding RNAs in bovine ovarian follicles and early embryos 

Dear Dr. Robert:

I'm pleased to inform you that your manuscript has been deemed suitable for publication in PLOS ONE. Congratulations! Your manuscript is now with our production department. 

Kind regards, 

on behalf of

Dr. Xiaoyong Sun 

Academic Editor

PLOS ONE